# The Single-Shore-Station-Based Position Estimation Method of an Automatic Identification System

**DOI:** 10.3390/s20061590

**Published:** 2020-03-12

**Authors:** Yi Jiang, Kai Zheng

**Affiliations:** 1Information Science and Technology College, Dalian Maritime University, Dalian 116026, China; 2Marine Electrical Engineering College, Dalian Maritime University, Dalian 116026, China; kzh@dlmu.edu.cn

**Keywords:** position estimation, ranging mode, single shore station, AIS

## Abstract

In order to overcome the vulnerability of the Global Navigation Satellite System (GNSS), the International Maritime Organization (IMO) initiated the ranging mode (R-Mode) of the automatic identification system (AIS) to provide resilient position data. As the existing AIS is a communication system, the number of shore stations as reference stations cannot satisfy positioning requirements. Especially in the area near a shore station, it is very common that a vessel can only receive signals from one shore station, where the traditional positioning method cannot be used. A novel position estimation method using multiple antennas on shipborne equipment is proposed here, which provides a vessel’s position even though the vessel can only receive signals from a single shore station. It is beneficial for solving positioning issues in proximity to the coast. Further, as the distances between different antennas to the shore station are not sufficiently independent, the positioning matrix can easily be near singularity or ill-conditioned; thus, an effective position solving method is derived. Furthermore, the proposed method is verified and evaluated in different scenarios by numerical simulation. We assessed the influencing factors of positioning performance, such as the vessel’s heading angle, the relative position, and the distances between the shore station and the vessel. The proposed method widely expands the application scope of the AIS R-Mode positioning system.

## 1. Introduction

The vulnerability of the Global Navigation Satellite System (GNSS) to both intentional and unintentional jamming and interference is an urgent problem to be solved in the e-Navigation Strategy Implementation Plan (SIP) of the International Maritime Organization (IMO) [1,2]. The automatic identification system (AIS) is widely used for maritime communication, and the IMO has mandated its installation since 2002 in order to avoid collisions [3,4,5,6]. By reusing the existing AIS infrastructure, the ranging mode (R-Mode) of AIS has been accepted as one of the alternative GNSS backup navigation systems of the future [7]. The resilient position data, which could be supplied by both satellite and terrestrial-based navigation systems, serve as the foundation for the e-Navigation SIP [8,9,10]. Its aim is to contribute to safe and reliable navigation at sea, especially for autonomous ship navigation.

Generally speaking, AIS is a very high frequency (VHF)-based communication system. AIS shore stations are the most critical components in a coastal AIS network. They can not only receive signals from shipborne equipment but also transmit signals within the coverage area. AIS R-Mode adds a ranging function without influencing the existing AIS communication capacity. The existing AIS shore stations are considered reference stations. A vessel can receive signals and derive ranging information to itself from shore stations, and as a consequence, the vessel’s position can be estimated [11,12]. It should be noted that the positional information in the existing AIS is now derived from GNSS [13]. If GNSS were to fail, the whole AIS would break down. Therefore, AIS R-Mode presents an efficient solution to overcome the dependence of AIS on GNSS. It will make AIS a comprehensive marine radio system integrated with communication and navigation for the e-navigation strategy. Therefore, many countries and scientists are researching AIS R-Mode. A European consortium of 12 research institutions, administrations, and industries in Germany, Poland, Sweden, and Norway has currently set up an R-Mode testbed in the Baltic Sea to study the feasibility of R-Mode [14,15]. The Maritime Safety Administration (MSA) and Dalian Maritime University have established the first AIS R-Mode testbed in China [11,16]. South Korea has performed some simulations of R-Mode with the integration of eLoran [17]. Further, the International Association of Marine Aids to Navigation and Lighthouse Authorities (IALA) is drafting the standard for R-Mode according to these countries’ research results [18].

In AIS R-Mode, shipborne AIS equipment receives VHF signals from AIS shore stations. If using traditional positioning methods in AIS R-Mode, such as the time of arrival (TOA) [19,20,21] or the time difference of arrival (TDOA) [22,23,24], the vessel needs to measure the signal transmission delay or relative delay. Then, the distance or distance difference can be obtained by multiplying it by the speed of light in the free space. Finally, the vessel’s position can be estimated according to the positioning equation. However, both methods require receiving signals from at least three different AIS shore stations. The existing AIS, though, is set up for communication without considering the requirement of positioning. Hence, a vessel cannot receive signals from three different AIS shore stations at the same time in some sea areas [25]. Therefore, the traditional positioning methods are not feasible. In this situation, if the vessel can receive signals from two AIS shore stations, a displacement correction position estimation method can be used to estimate the vessel’s position [26,27]. The principle of this method is to calculate the displacement of the vessel for a period of time according to parameters, such as the heading and speed, provided by auxiliary sensors. The relationship of positional information between the adjacent moments can be derived by the displacement vector. Finally, the vessel’s position is estimated by the continuous range measurements in adjacent moments. However, when a vessel can only receive signals from one shore station, the existing position estimation methods cannot be utilized. 

We have proposed a position estimation method for AIS R-Mode, especially in the area close to a shore station. This method aims to estimate the vessel’s position when it can only receive signals from a single AIS shore station. The contributions of this study are as follows.
Traditional positioning in AIS R-Mode needs at least three reference stations. This paper proposes a position estimation method for a single shore station by using multiple antennas on a vessel.Due to the limited size of the vessel, the distances between different antennas to the shore station are approximately the same and not sufficiently independent. The positioning matrix is prone to being near singularity or ill-conditioned. The second significant finding is an effective position solving method for this near-singular positioning matrix.The proposed method was verified and evaluated in different scenarios using MATLAB simulations. The positioning performance of the proposed method was found to be influenced by the relative position between the antennas and the shore station. Further, position errors increased as the distance increased.

The remainder of this paper is organized as follows: The architecture of the AIS R-Mode positioning system is given, and different position estimation methods that can be used in AIS R-Mode are discussed in Section 2. In Section 3, we present a novel position estimation method for when a vessel can only receive signals from a single AIS shore station. Furthermore, an effective position solving method to avoid singularity is derived for the conditions of three antennas and more than three antennas. Then, four different simulation scenarios are introduced to verify and evaluate the proposed method in Section 4. Simulation results are analyzed, and the influencing factors of positioning performance are discussed in Section 5. Finally, conclusions are put forth in Section 6.

## 2. AIS R-Mode Positioning

The AIS R-mode positioning system is comprised of AIS shore stations and vessels. The system framework is illustrated in Figure 1. The AIS shore stations, as positioning reference stations, transmit VHF ranging signals periodically. The circles in Figure 1 indicate the coverage areas of shore stations. Only the vessels in the coverage area of the shore station can receive its signals. Then, the vessel estimates its position according to its signals received from shore stations.

In Figure 1, if the vessel, such as Vessel 1, can receive signals from at least three AIS shore stations, either the TOA method or the TDOA method can be utilized to estimate the vessel’s position in AIS R-mode positioning [25]. However, the layout of existing AIS shore stations was originally designed to satisfy communication requirements. A single shore station is sufficient for the vessel to communicate. However, as the AIS traffic load increased, more shore stations were established to increase the signal coverage rate. Still, areas exist where there are insufficient numbers of shore stations that can act as the reference stations for positioning [25]. For instance, some vessels can only receive signals from two shore stations, such as Vessel 2 in Figure 1. In this situation, the TOA method can be enhanced with the displacement correction so that the vessel’s position can be estimated [26]. However, some vessels can only receive signals from a single shore station, shown in Figure 1, such as Vessel 3. Previous studies have based their criteria for at least two shore stations, so they were not able to estimate vessels’ positions in this condition. This paper mainly focuses on the position estimation method based on a single shore station in AIS R-Mode.

## 3. Positioning Method for a Single Shore Station

The proposed positioning method based on a single shore station in AIS R-Mode uses multiple antennas on a vessel. Figure 2 presents the global coordinate system for R-Mode positioning and the local coordinate system for the proposed positioning method. *O* is the origin of the global system, the *Y*-axis is directed towards the north, and the *X*-axis is perpendicular to the *X*-axis. (*X*, *Y*) indicates the coordination in the global system. In Figure 2, *B* and *M* denote the shore station and the vessel, respectively. An inverted triangle depicts an antenna on the vessel. Multiple antennas on the vessel are shown in Figure 2. The global coordinates of the shore station *B* are (*X_B_, Y_B_*). (*X_i_, Y_i_*) are the global position coordinates of the *i*th antenna of the vessel. In the local coordinate system, *o* is the origin, and it is the center of the vessel; the *x*-axis is directed towards the heading direction, and the *y*-axis is perpendicular to the *x*-axis. (*x_i_*, *y_i_*) are the position coordinates of the *i*th antenna on the vessel in the local coordinate system.

The distance between the reference shore station and the *i*th antenna on the vessel can be calculated using the following simple equation [28].
(1)$${\bar{R}}_{i}=c\left(T_{r_{i}}-T_{t}\right)$$
where *c* is the speed of light in the free space; *T_t_* is the signal transmission time, which can be obtained according to the AIS VHF signal from the shore station; and $T_{r_{i}}$ is the signal arrival time for the *i*th antenna of vessel *M*, which can be obtained by the vessel. 

However, as time synchronization between a vessel and shore stations is a difficult task to achieve in reality, ${\bar{R}}_{i}$ is the measured distance between the *i*th antenna and the shore station *B*, not equal to the actual distance *R_i_*. The positioning equation between the *i*th antenna of the vessel and the shore station is
(2)$${\bar{R}}_{i}=R_{i}\left(X_{i},Y_{i}\right)+c\Delta T$$
where subscript *i* represents the *i*th antenna, and Δ*T* is the clock offset between *M* and *B*. The accurate distance *R_i_* can be calculated by the Euclidean distance formula:(3)$$R_{i}\left(X_{i},Y_{i}\right)={\left({\left(X_{i}-X_{B}\right)}^{2}+{\left(Y_{i}-Y_{B}\right)}^{2}\right)}^{\frac{1}{2}}$$

According to Equation (3), Equation (2) can be further written as
(4)$${\bar{R}}_{i}={\left({\left(X_{i}-X_{B}\right)}^{2}+{\left(Y_{i}-Y_{B}\right)}^{2}\right)}^{\frac{1}{2}}+c\Delta T$$

If one antenna is installed at the center of the vessel, the position coordinates of the vessel can be represented as (*X*_0_*, Y*_0_).

The heading angle *θ* of the vessel at any time can be obtained according to the outputs of shipborne equipment, including a magnetic compass, a gyrocompass, and so forth. In the global coordinate system, the coordinates of the *i*th antenna (*X_i_*, *Y_i_*) can be expressed as
(5)$$\left[\begin{array}{c} X_{i}\\ Y_{i} \end{array}\right]=\left[\begin{array}{c} X_{0}\\ Y_{0} \end{array}\right]+T\left(\theta \right)\left[\begin{array}{c} x_{i}\\ y_{i} \end{array}\right]$$
where *T*(*θ*) is called the rotation matrix and is given below:(6)$$T\left(\theta \right)=\left[\begin{array}{cc} -\sin\theta & \cos\theta \\ \cos\theta & \sin\theta \end{array}\right]$$
substituting Equation (5) into Equation (3), we have
(7)$$R_{i}\left(X_{i},Y_{i}\right)=R_{i}^{\prime }\left(X_{0},Y_{0}\right)$$

The positioning equation of (*X*_0_, *Y*_0_) can be written as
(8)$${\bar{R}}_{i}=R_{i}^{\prime }\left(X_{0},Y_{0}\right)+c\Delta T$$

As Equation (8) is a nonlinear equation, the first step of solving the positioning equation is linearization. A Taylor series is used, and the first-order terms are retained:(9)$${\bar{R}}_{i}-R_{i}^{\prime }\left({\hat{X}}_{0},{\hat{Y}}_{0}\right)-c\Delta \hat{T}={\frac{\partial R_{i}^{\prime }}{\partial X_{0}}|}_{\left({\hat{X}}_{0},{\hat{Y}}_{0}\right)}\delta X_{0}+{\frac{\partial R_{i}^{\prime }}{\partial Y_{0}}|}_{\left({\hat{X}}_{0},{\hat{Y}}_{0}\right)}\delta Y_{0}+c\delta T$$
where (${\hat{X}}_{0}$, ${\hat{Y}}_{0}$) is the initial estimated coordinate of *M*, and (*δX*_0_, *δY*_0_, *δT)* are the corrections of the corresponding estimated values. 

However, due to the limitation of the vessel size, the distances between different antennas to the shore station are approximately the same and not sufficiently independent. Therefore, the positioning matrix given by Equation (9) easily becomes a near-singular matrix or an ill-conditioned matrix, which is difficult to solve. Thus, the Taylor series expansion method [29,30] or the least-squares method [31,32] is widely used in position estimation. Both of these methods require good initial values; otherwise, it is difficult for the solution to achieve convergence [33]. Therefore, they are not suitable for solving the position estimation situation proposed in this paper. The Chan method is a TDOA-based localization algorithm, which can provide a closed-form solution for arbitrarily placed reference nodes [34]. Inspired by the Chan method, we solved the proposed positioning equation as follows:

According to Equations (7) and (8), we rewrote Equation (8) as
(10)$$R_{i}\left(X_{i},Y_{i}\right)=R_{i}^{\prime }\left(X_{0},Y_{0}\right)={\bar{R}}_{i0}+R_{0}\left(X_{0},Y_{0}\right)$$
where
(11)$${\bar{R}}_{i0}={\bar{R}}_{i}-{\bar{R}}_{0}$$
Substituting Equation (10) into Equation (3), we obtained
(12)$${\bar{R}}_{i0}^{2}+2{\bar{R}}_{i0}R_{0}+R_{0}{}^{2}=X_{i}^{2}+Y_{i}^{2}+X_{B}^{2}+Y_{B}^{2}-2\left(X_{i}X_{B}+Y_{i}Y_{B}\right)$$
If *i* = 0 in Equation (3), it can be simplified as
(13)$$R_{0}{}^{2}=X_{0}^{2}+Y_{0}^{2}+X_{B}^{2}+Y_{B}^{2}-2\left(X_{0}X_{B}+Y_{0}Y_{B}\right)$$

Then, subtracting Equation (13) from Equation (12), the result is
(14)$${\bar{R}}_{i0}^{2}+2{\bar{R}}_{i0}R_{0}=X_{i}^{2}+Y_{i}^{2}-\left(X_{0}^{2}+Y_{0}^{2}\right)-2\left(X_{i0}X_{B}+Y_{i0}Y_{B}\right)$$
according to
(15)$$X_{i}^{2}+Y_{i}^{2}={\left[\begin{array}{c} X_{0}\\ Y_{0} \end{array}\right]}^{T}\left[\begin{array}{c} X_{0}\\ Y_{0} \end{array}\right]+2{\left[\begin{array}{c} x_{i}\\ y_{i} \end{array}\right]}^{T}T\left(\theta \right)\left[\begin{array}{c} X_{0}\\ Y_{0} \end{array}\right]+{\left[\begin{array}{c} x_{i}\\ y_{i} \end{array}\right]}^{T}\left[\begin{array}{c} x_{i}\\ y_{i} \end{array}\right]$$
Substituting Equations (5) and (15) into Equation (14), we obtained
(16)$${\bar{R}}_{i0}^{2}+2{\bar{R}}_{i0}R_{0}=2{\left[\begin{array}{c} x_{i}\\ y_{i} \end{array}\right]}^{T}T\left(\theta \right)\left[\begin{array}{c} X_{0}\\ Y_{0} \end{array}\right]+{\left[\begin{array}{c} x_{i}\\ y_{i} \end{array}\right]}^{T}\left[\begin{array}{c} x_{i}\\ y_{i} \end{array}\right]-2{\left[\begin{array}{c} x_{i}\\ y_{i} \end{array}\right]}^{T}T\left(\theta \right)\left[\begin{array}{c} X_{B}\\ Y_{B} \end{array}\right]$$

### 3.1. Condition of Three Antennas

If there are three antennas on the vessel, according to Equation (16), the positioning matrix can be written simply as
(17)$$\left[\begin{array}{c} X_{0}\\ Y_{0} \end{array}\right]={\left({\left[\begin{array}{cc} \begin{array}{c} x_{1}\\ y_{1} \end{array} & \begin{array}{c} x_{2}\\ y_{2} \end{array} \end{array}\right]}^{T}T\left(\theta \right)\right)}^{-1}\left\{\frac{1}{2}\left(\left[\begin{array}{c} {\bar{R}}_{10}^{2}\\ {\bar{R}}_{20}^{2} \end{array}\right]-\left[\begin{array}{c} \left[\begin{array}{cc} x_{1} & y_{1} \end{array}\right]{\left[\begin{array}{cc} x_{1} & y_{1} \end{array}\right]}^{T}\\ \left[\begin{array}{cc} x_{2} & y_{2} \end{array}\right]{\left[\begin{array}{cc} x_{2} & y_{2} \end{array}\right]}^{T} \end{array}\right]\right)+\left[\begin{array}{c} {\bar{R}}_{10}\\ {\bar{R}}_{20} \end{array}\right]R_{0}\right\}+\left[\begin{array}{c} X_{B}\\ Y_{B} \end{array}\right]$$
where *R*_0_ can be expressed by (*X*_0_, *Y*_0_), given by
(18)$$R_{0}^{2}={\left[\begin{array}{c} X_{0}-X_{B}\\ Y_{0}-Y_{B} \end{array}\right]}^{T}\left[\begin{array}{c} X_{0}-X_{B}\\ Y_{0}-Y_{B} \end{array}\right]$$

The equation can be solved by substituting Equation (17) into Equation (18), and we can calculate the vessel position coordinates (*X*_0_, *Y*_0_) according to Equation (17).

### 3.2. Condition of More Than Three Antennas

If there are *M* antennas on the vessel and *M* > 3, according to Equation (16), the positioning matrix can be written as
(19)$${\left[\begin{array}{c} x_{i}\\ y_{i} \end{array}\right]}^{T}T\left(\theta \right)\left[\begin{array}{c} X_{0}\\ Y_{0} \end{array}\right]-{\bar{R}}_{i0}R_{0}=\frac{1}{2}\left({\bar{R}}_{i0}^{2}-{\left[\begin{array}{c} x_{i}\\ y_{i} \end{array}\right]}^{T}\left[\begin{array}{c} x_{i}\\ y_{i} \end{array}\right]\right)+{\left[\begin{array}{c} x_{i}\\ y_{i} \end{array}\right]}^{T}T\left(\theta \right)\left[\begin{array}{c} X_{B}\\ Y_{B} \end{array}\right]$$
Substituting Equation (6) into Equation (19), we obtained
(20)$${\left[\begin{array}{c} \left(\begin{array}{cc} -\sin\theta & \cos\theta \end{array}\right){\left[\begin{array}{cc} x_{i} & y_{i} \end{array}\right]}^{T}\\ \left(\begin{array}{cc} \cos\theta & \sin\theta \end{array}\right){\left[\begin{array}{cc} x_{i} & y_{i} \end{array}\right]}^{T}\\ -{\bar{R}}_{i0} \end{array}\right]}^{T}\left[\begin{array}{c} \begin{array}{c} X_{0}\\ Y_{0} \end{array}\\ R_{0} \end{array}\right]=\frac{1}{2}\left({\bar{R}}_{i0}^{2}-{\left[\begin{array}{c} x_{i}\\ y_{i} \end{array}\right]}^{T}\left[\begin{array}{c} x_{i}\\ y_{i} \end{array}\right]\right)+{\left[\begin{array}{c} x_{i}\\ y_{i} \end{array}\right]}^{T}T\left(\theta \right)\left[\begin{array}{c} X_{B}\\ Y_{B} \end{array}\right]$$
The positioning matrix can be simplified as
(21)HX=b
where
(22)$$X={\left[\begin{array}{ccc} X_{0} & Y_{0} & R_{0} \end{array}\right]}^{T}$$
(23)$$H=\left[\begin{array}{c} \begin{array}{ccc} \left(\begin{array}{cc} -\sin\theta & \cos\theta \end{array}\right)\left[\begin{array}{c} x_{1}\\ y_{1} \end{array}\right] & \left(\begin{array}{cc} \cos\theta & \sin\theta \end{array}\right)\left[\begin{array}{c} x_{1}\\ y_{1} \end{array}\right] & -{\bar{R}}_{10} \end{array}\\ \begin{array}{ccc} \left(\begin{array}{cc} -\sin\theta & \cos\theta \end{array}\right)\left[\begin{array}{c} x_{2}\\ y_{2} \end{array}\right] & \left(\begin{array}{cc} \cos\theta & \sin\theta \end{array}\right)\left[\begin{array}{c} x_{2}\\ y_{2} \end{array}\right] & -{\bar{R}}_{20} \end{array}\\ \begin{array}{ccccccccc} & & & & & \vdots & & & \begin{array}{ccccccccc} \begin{array}{ccccccccc} & & & & & & & \vdots & \end{array} & & & & & & & & \vdots \end{array} \end{array}\\ \begin{array}{ccc} \left(\begin{array}{cc} -\sin\theta & \cos\theta \end{array}\right)\left[\begin{array}{c} x_{M-1}\\ y_{M-1} \end{array}\right] & \left(\begin{array}{cc} \cos\theta & \sin\theta \end{array}\right)\left[\begin{array}{c} x_{M-1}\\ y_{M-1} \end{array}\right] & -{\bar{R}}_{\left(M-1\right)0} \end{array} \end{array}\right]$$
(24)$$b=\left[\begin{array}{c} \frac{1}{2}\left({\bar{R}}_{10}^{2}-{\left[\begin{array}{c} x_{1}\\ y_{1} \end{array}\right]}^{T}\left[\begin{array}{c} x_{1}\\ y_{1} \end{array}\right]\right)+{\left[\begin{array}{c} x_{1}\\ y_{1} \end{array}\right]}^{T}T\left(\theta \right)\left[\begin{array}{c} X_{B}\\ Y_{B} \end{array}\right]\\ \frac{1}{2}\left({\bar{R}}_{20}^{2}-{\left[\begin{array}{c} x_{2}\\ y_{2} \end{array}\right]}^{T}\left[\begin{array}{c} x_{2}\\ y_{2} \end{array}\right]\right)+{\left[\begin{array}{c} x_{2}\\ y_{2} \end{array}\right]}^{T}T\left(\theta \right)\left[\begin{array}{c} X_{B}\\ Y_{B} \end{array}\right]\\ \vdots \\ \frac{1}{2}\left({\bar{R}}_{\left(M-1\right)0}^{2}-{\left[\begin{array}{c} x_{M-1}\\ y_{M-1} \end{array}\right]}^{T}\left[\begin{array}{c} x_{M-1}\\ y_{M-1} \end{array}\right]\right)+{\left[\begin{array}{c} x_{M-1}\\ y_{M-1} \end{array}\right]}^{T}T\left(\theta \right)\left[\begin{array}{c} X_{B}\\ Y_{B} \end{array}\right] \end{array}\right]$$

As measurement errors exist in ${\bar{R}}_{i0}$, the positioning matrix of Equation (21) was rewritten as
(25)ψ=b−HX
where
(26)$$\psi_{i}=R_{i}cn_{i}+0.5c^{2}n_{i}^{2}$$
where *n_i_* is an error corresponding to ${\bar{R}}_{i0}$. Due to *R_i_* ≈ *R*_0_, **R** ≈ *R*_0_**I**. **I** is an identity matrix. Equation (25) can be written as
(27)$$\psi \approx cR_{0}n+0.5c^{2}n⊙n$$

Therefore, solving the positioning matrix requires actually finding the value of **X** where ||**b** – **HX**|| is the minimum. Using the weighted least-squares method, the solution of Equation (25) is
(28)$$X={\left(H^{T}Q^{-1}H\right)}^{-1}H^{T}Q^{-1}b$$
where **Q** is the covariance matrix of the measurement error. The covariance matrix of **X** can be expressed as
(29)$$\mathrm{cov}\left(X\right)\approx c^{2}R_{0}^{2}{\left(H^{T}Q^{-1}H\right)}^{-1}$$

In order to improve the accuracy, assuming that the elements of **X** are independent, we defined
(30)$$X=\left[\begin{array}{c} {\tilde{X}}_{0}+e_{X}\\ {\tilde{Y}}_{0}+e_{Y}\\ {\tilde{R}}_{0}+e_{R} \end{array}\right]$$
where **e** represents the estimation errors of **X**. Subtracting the first two components of **X** by (*X_B_*, *Y_B_*), we obtained
(31)φ=h−GZ
where
(32)$$Z=\left[\begin{array}{c} {\left({\tilde{X}}_{0}-X_{B}\right)}^{2}\\ {\left({\tilde{Y}}_{0}-Y_{B}\right)}^{2} \end{array}\right]$$
(33)$$G=\left[\begin{array}{cc} 1 & 0\\ 0 & 1\\ 1 & 1 \end{array}\right]$$
(34)$$h=\left[\begin{array}{c} {\left(X_{0}-X_{B}\right)}^{2}\\ {\left(Y_{0}-Y_{B}\right)}^{2}\\ R_{0}^{2} \end{array}\right]$$

The solution of Equation (31) is
(35)$$Z={\left(G^{T}\Phi^{-1}G\right)}^{-1}G^{T}\Phi^{-1}h$$
where
(36)$$\Phi =E\left(\varphi^{T}\varphi \right)$$
we know that
(37)$$\begin{array}{l} \varphi_{1}=2\left(X_{0}-X_{B}\right)e_{X}+e_{X}^{2}\approx 2\left(X_{0}-X_{B}\right)e_{X}\\ \varphi_{2}=2\left(Y_{0}-Y_{B}\right)e_{Y}+e_{Y}^{2}\approx 2\left(Y_{0}-Y_{B}\right)e_{Y}\\ \varphi_{3}=2R_{0}e_{R}+e_{R}^{2}\approx 2R_{0}e_{R} \end{array}\}$$
Equation (36) can be derived as
(38)Φ=4Dcov(X)D
where
(39)$$D=\mathrm{diag}\left\{\left(X_{0}-X_{B}\right),\left(Y_{0}-Y_{B}\right),R_{0}\right\}$$

Substituting Equation (28) into Equation (37),
(40)$$\Phi =E\left(\varphi^{T}\varphi \right)=4D\mathrm{cov}\left(X\right)D=4c^{2}R_{0}{}^{2}D{\left(H^{0}{}^{T}Q^{-1}H^{0}\right)}^{-1}D$$

Similarly, substituting Equation (38) into Equation (34),
(41)$$Z={\left(G^{T}D^{-1}H^{0}{}^{T}Q^{-1}H^{0}D^{-1}G\right)}^{-1}G^{T}D^{-1}H^{0}{}^{T}Q^{-1}H^{0}D^{-1}h$$

The covariance of **Z** is
(42)$$\mathrm{cov}\left(Z\right)={\left(G^{T}\Phi^{-1}G\right)}^{-1}$$

Finally, the vessel position is estimated as
(43)$$X^{\prime }=\pm \sqrt{Z}+\left[\begin{array}{c} X_{B}\\ Y_{B} \end{array}\right]$$

According to the approximate position of the vessel, the minus sign or the plus sign can be determined in Equation (43). The covariance matrix of position estimation **X’**, corresponding to positioning precision, is given by
(44)$$\mathrm{cov}\left(X^{\prime }\right)=\frac{1}{4}B^{-1}\mathrm{cov}\left(Z\right)B^{-1}\approx c^{2}R_{0}{}^{2}{\left(BG^{T}D^{-1}\left(H^{0}{}^{T}Q^{-1}H^{0}\right)D^{-1}GB\right)}^{-1}$$
where
(45)$$B=\mathrm{diag}\left\{\left(X_{0}-X_{B}\right),\left(Y_{0}-Y_{B}\right)\right\}$$

The position root-mean-square error (RMSE) is equal to the root of the sum of the diagonal elements of cov(**X’**). Furthermore, to further improve the accuracy of the estimated position, smooth filtering was also used based on the estimated position **X’** of Equation (43). We formulated this smooth problem in MATLAB by simply using the smoothdata function to smooth noisy position data.

## 4. Simulation Scenario

In order to verify the proposed position estimation method for a single shore station in AIS R-Mode, numerical simulation of positioning errors was performed in different scenarios using MATLAB. A real AIS shore station named Huangbaizui was used, the latitude of which is 38°54.2850′ N, and the longitude is 121°42.9500′ E, as shown in Figure 3a. The location of the vessel was very close to the Huangbaizui shore station, where the vessel could only receive signals from the Huangbaizui shore station. Its initial latitude was 38°40.1690′ N, and the longitude was 122°10.1020′ E. We used a Gauss–Krüger projection with six-degree zones [35]; the shore station coordinates (*X_B_, Y_B_*) were [4.3087 × 10^6^, −1.1139 × 10^5^]; and the vessel’s initial position (*X*_0_*, Y*_0_) was [4.3092 × 10^6^, −1.0938 × 10^5^]. The unit of measurement was meters. Figure 3b shows the geometric relationship between the shore station and the antennas of the vessel when the heading angle of the vessel was 0°. The blue star denotes the position of the AIS shore station. The red asterisk is the position of the vessel’s center. The green crosses denote the other three antennas on the vessel. As the antennas’ positions were linked with the vessel’s dimensions, we introduced a common engineering vessel, named Maochang 526, to set up the antennas’ location parameters. As Maochang 526′s cargo was 650 t, the length was 68 m, and the width was 14 m, the position coordinates of the four antennas were [0, 0], [15, 7], [30, 0], [15, −7] in the local coordinate system of the vessel, as shown in Figure 3b. 

In order to assess the performance of the proposed positioning method for a single shore station, positioning errors were evaluated in this study in the following four simulation scenarios.
Scenario 1: The heading angle was 0°. In order to evaluate the effects of the heading angle of the vessel on the positioning errors, the location of the vessel was fixed and just the heading angle was changing.Scenario 2: The vessel moved along a circular trajectory, the center of which was the location of the shore station. During the movement, the heading angle of the vessel was constant.Scenario 3: The vessel moved along a circular trajectory, the center of which was the location of the shore station. During the movement, the heading angle of the vessel was continuously adjusted to maintain the relative position between the antennas and the shore station.Scenario 4: The vessel moved toward or away from the shore station with a constant heading angle. Only the distance between the shore station and the vessel was changing.

To bring simulation closer to reality, we used the Monte Carlo method to create measurement noises [36]. We ran this 1000 times at every vessel position in the above scenarios by adding Gaussian random noise with a mean root square of 0.001.

## 5. Simulation Results Analysis

### 5.1. Scenario 1: Positioning Errors Vary with Heading Angles

In Scenario 1, the position of the vessel was fixed. Further, the heading angle of the vessel was increased from 0° to 90°. Figure 4 shows the antenna locations of the vessel with different heading angles *θ*. The red cross in Figure 4 denotes the vessel’s location—the center of the vessel. The green quadrilaterals represent the layout of the antennas in the vessel with different heading angles.

The theoretical RMSE of the proposed method in this study at different heading angles is given in Table 1. Moreover, the calculated RMSE and the average positioning error after smooth filtering according to the simulation results in Scenario 1 are also given.

From Table 1, it can be seen clearly that the positioning errors increased as the heading angles increased. The reason for the increase of the RMSE was that the correlation between distances from different antennas to the shore station was increasing with the relative position change between the antennas and the shore station. These results indicate that the calculated RMSE based on the simulation positioning results is consistent with the RMSE according to the theoretical derivation. In addition, positioning errors could be reduced effectively by using the smoothdata function in MATLAB.

### 5.2. Scenario 2: Positioning Errors Vary with Vessel’s Position

In Scenario 2, the vessel navigated around the shore station. The relative azimuth angle *η* shown in Figure 2 and Figure 5 is the angle between the north vector and the shore to the vessel vector. It started at 5° and stopped every 10° to stabilize its heading angle on 0° and get its position. The vessel continued to navigate until *η* became 85°. The shore station was the center of the motion curve. Figure 5 shows the positioning results of the vessel as its position changed. The red cross is the actual position of the vessel. The black asterisk is the estimated position using the proposed method. The vertexes of the green diamond represent the positions of antennas.

*η* is the relative azimuth angle of the shore station shown in Figure 2 and Figure 5. Table 2 shows the positioning results when the vessel stopped every 10° until *η* became 85° in Scenario 2.

It can be seen from Figure 5 and Table 2 that the RMSE was changing with the relative position between the antennas and the shore station. The RMSE was largest and the positioning result was the worst when the relative azimuth angle between the vessel and the shore station was 25°. In this situation, the distances between different antennas and the shore station were not sufficiently independent. Antennas ② and ④, and the shore station, were almost on the same line, shown in Figure 5. Therefore, the positioning matrix attained singularity in this situation. The calculated RMSE based on the simulation positioning results was much larger than the theoretical RMSE, as the noise had a significant impact on the positioning matrix. Even using a smoothing filter could not reduce the position errors.

### 5.3. Scenario 3: Positioning Errors Vary with Relative Position

In Scenario 3, the vessel navigated around the shore station, and the shore station was still the center of the motion curve. It stopped every 10° to adjust its heading angle to maintain the relative positional relationship between the antennas and the shore station, and get its position. The positioning results of the vessel are shown in Figure 6. We can see that the relative position between the antennas and the shore station remained the same. In addition, the position errors were similar, although the vessel was in different locations.

The numerical positioning results in Scenario 3 are given in Table 3.

According to the simulation results shown in Table 3, the theoretical RMSE was the same. The reason for this was that the relative position between the antennas and the shore station was the same. Further, the calculated RMSE was almost the same. Positioning errors by the smoothing filter were reduced and were at the same level, substantially, when the vessel was in different positions. In Scenario 1, the heading angles of the vessel were changing. In Scenario 2, the position of the vessel was changing. However, in this scenario, the relative position was the same, although both the heading angle and the position were changing. Therefore, it can be concluded that, as long as the distance is constant and the relative positional relationship remains unchanged, the positioning performance is the same, regardless of whether the position and the heading angles of the vessel are changing.

### 5.4. Scenario 4: Positioning Errors Vary with Distances

In Scenario 4, the vessel moved toward or away from the shore station. The distance between the shore station and the vessel was changing from 100 to 1000 m. During the movement, the heading angle of the vessel was always 0°. Figure 7 shows the estimated positioning results as the distance changed. Position errors varied with the distance, as depicted in Figure 8.

Table 4 shows the positioning results in Scenario 4. It can be observed that the simulated RMSE and position errors increased as the distance increased. Therefore, the performance was better in the area closer to the shore station. Correspondingly, the proposed method is precisely used for positioning based on a single shore station, which is suitable for areas closer to the shore station.

## 6. Conclusions

As the existing AIS is a communication system, when AIS shore stations are used as positioning reference stations in AIS R-Mode, a vessel can only receive signals from a single shore station in some areas, especially in an area close to the shore station. In this situation, traditional positioning methods are not suitable. A position estimation method using multiple antennas was proposed in this paper to solve this problem. The geometric relationship between the antennas in the local coordinate system is converted into the global positioning coordinate system using the heading angle. Then, the positioning matrix is obtained according to measurements from multiple antennas. However, due to the size of the vessel, the distances between different antennas to the shore station are not sufficiently independent. Therefore, the positioning matrix of this proposed method is easily near singularity or ill-conditioned. A novel method to solve the positioning matrix was presented here, rather than the Taylor series expansion method, which can effectively prevent singularity. Finally, we verified the validity of the proposed method in diverse scenarios by numerical simulations. The influencing factors of positioning performance were analyzed, such as heading angle, relative position, and distances. The positioning performance became worse as the distance increased. Fortunately, the proposed method was exactly suited for the area close to the shore station, where the vessel can only receive signals from this shore station. Further, the positioning errors were mainly affected by the relative positional relationship between the antennas and the shore station. As it is not necessary to establish new AIS shore stations, the proposed method can help the AIS R-Mode positioning system to expand its application scope. A possible area of future research would be to investigate the improved position estimation method based on a single shore station, which can reduce positioning errors in the far area of the shore station. This would help to further expand AIS R-Mode applications.

## Figures and Tables

**Figure 1 sensors-20-01590-f001:**
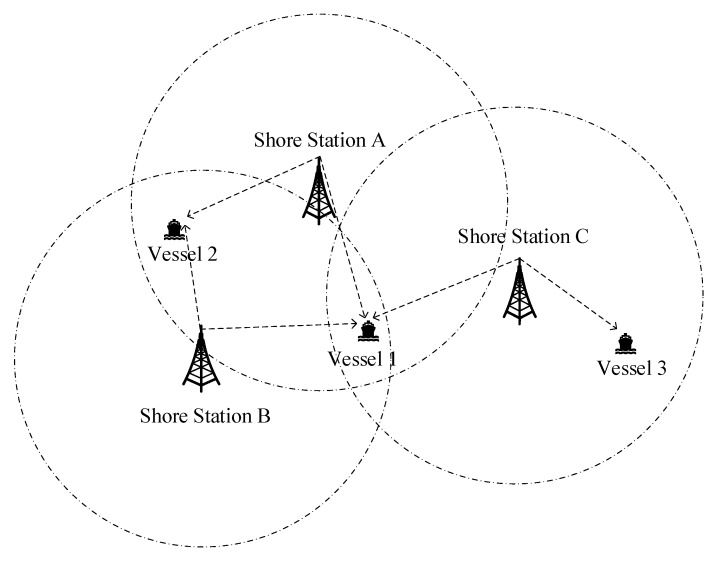
Coverage areas of automatic identification system (AIS) stations: different cases.

**Figure 2 sensors-20-01590-f002:**
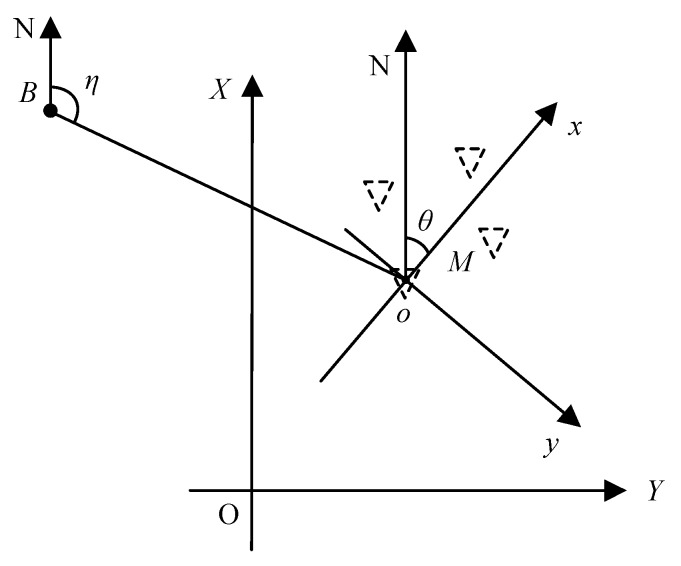
Global and local coordinate systems for positioning.

**Figure 3 sensors-20-01590-f003:**
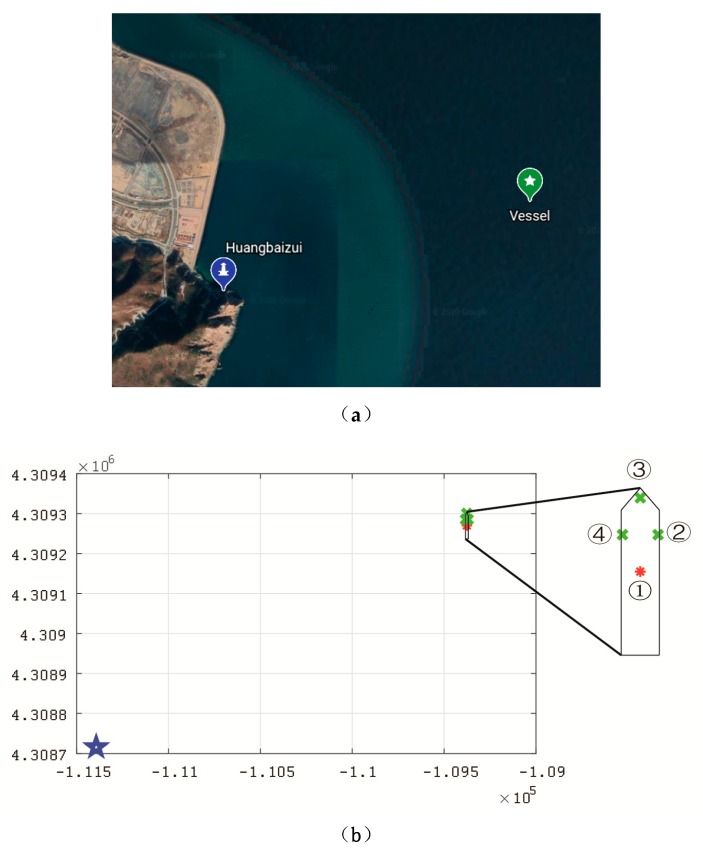
Distribution of single shore station and initial vessel location. (**a**) distribution on google map (**b**) distribution using a Gauss–Krüger projection.

**Figure 4 sensors-20-01590-f004:**
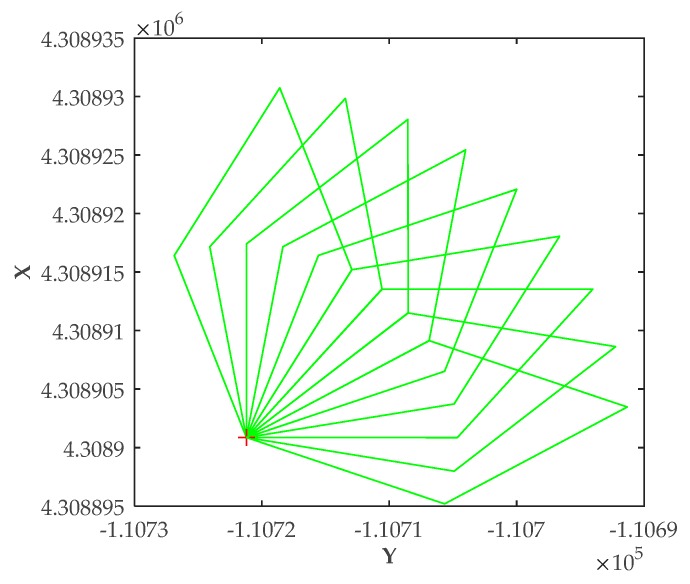
Antenna layout with different heading angles.

**Figure 5 sensors-20-01590-f005:**
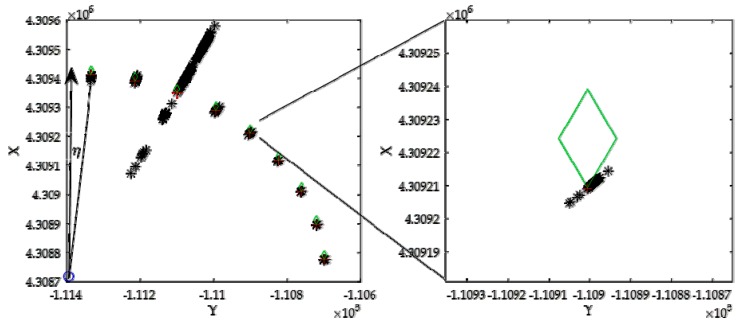
Positioning results at different locations.

**Figure 6 sensors-20-01590-f006:**
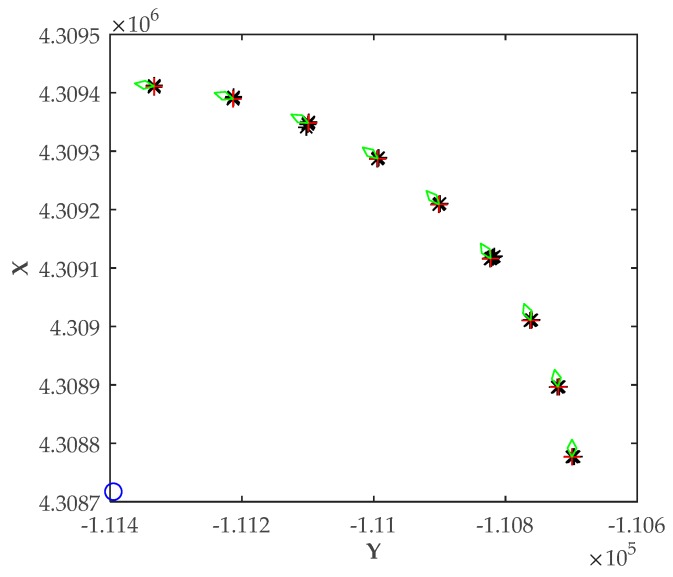
Positioning results with unchanged relative position.

**Figure 7 sensors-20-01590-f007:**
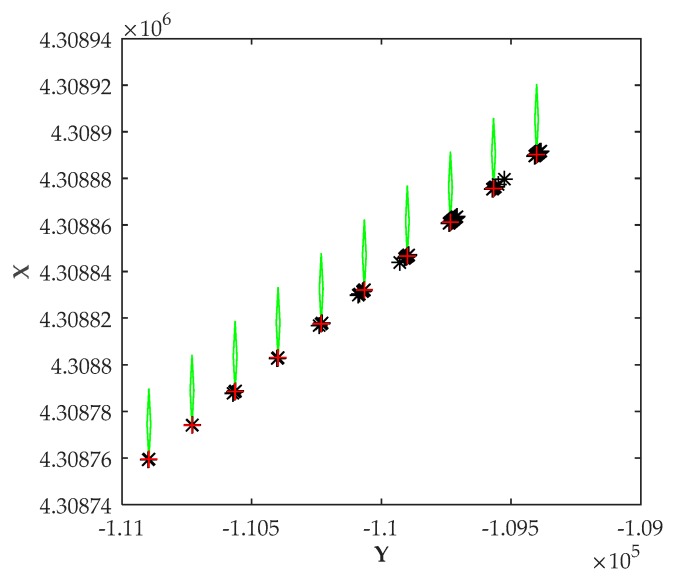
Positioning results with changing distance.

**Figure 8 sensors-20-01590-f008:**
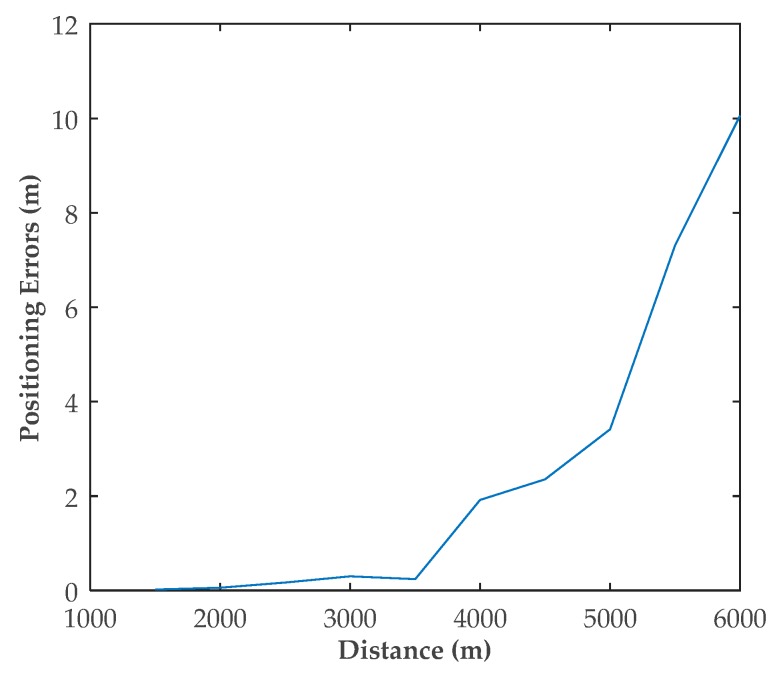
Positioning results at different distances.

**Table 1 sensors-20-01590-t001:** Positioning results of Scenario 1.

Heading Angle *θ*	5	15	25	35	45	55	65	75	85
Theoretical RMSE	1.609	2.849	4.439	6.381	8.682	11.324	14.334	17.685	21.350
Simulated RMSE	1.609	2.888	4.441	6.386	8.683	11.332	14.334	17.688	21.395
Filtered position error	0.019	0.057	0.165	0.240	0.298	1.916	2.354	3.412	7.312

**Table 2 sensors-20-01590-t002:** Positioning results of Scenario 2.

Relative Azimuth Angle *η*	85	75	65	55	45	35	25	15	5
Theoretical RMSE	3.117	3.418	4.077	5.412	8.451	18.688	271.811	22.850	15.274
Simulated RMSE	3.118	3.420	4.078	5.408	8.473	18.690	352.870	22.950	15.276
Filtered position error	0.092	0.102	0.092	0.231	1.464	3.797	588.233	7.592	3.400

**Table 3 sensors-20-01590-t003:** Positioning results of Scenario 3.

Heading Angle *θ*	85	75	65	55	45	35	25	15	5
Theoretical RMSE	3.117	3.117	3.117	3.117	3.117	3.117	3.117	3.117	3.117
Simulated RMSE	3.119	3.115	3.119	3.118	3.113	3.117	3.118	3.116	3.117
Filtered position error	0.142	0.077	0.142	0.141	0.233	0.041	0.245	0.072	0.031

**Table 4 sensors-20-01590-t004:** Positioning result of Scenario 4.

Distance	1500	2000	2500	3000	3500	4000	4500	5000	5500	6000
Theoretical RMSE	1.609	2.849	4.441	6.385	8.682	11.332	14.334	17.688	21.350	25.422
Simulated RMSE	1.609	2.849	4.449	6.386	8.683	11.334	14.334	17.689	21.395	25.454
Filtered position error	0.019	0.057	0.1654	0.298	0.4202	1.916	2.354	3.412	7.312	10.057

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
