# Peer review of "The Single-Shore-Station-Based Position Estimation Method of an Automatic Identification System"

_sensors, 2020, doi:10.3390/s20061590_

Round 1

Reviewer 1 Report

The article deals with estimation position method with help single shore station based on the Automatic Identification System (AIS).

The existing AIS is a communication system for maritime information transition, such as position, course, and speed. AIS base stations have been already located along coastlines of International Maritime Organizations (IMO’s) member countries, and shipborne AIS equipment is mandatory on most vessels according to the IMO requirements. AIS was introduced as a ship-to-ship and ship-to-shore reporting system intended to increase the safety of life at sea and to improve control and monitoring of maritime traffic. If using the traditional positioning method in AIS, such as the time of arrival (TOA) and time difference of arrival (TDOA) the vessel need to measure distance from at least three different AIS shore stations and then the position can be estimated. The authors suggest instead of the three different AIS shore stations, use one AIS shore station only and the vessel equipped with more antennas. The authors derivate the positioning matrix for case three antennas and case for more than three ones and made more numerical simulations with results that the  positioning performance becomes worse as the distance increasing.

Within the conclusion authors recommend suggested method namely for the situation when the vessel is in the nearer area of the shore station.

Some remarks:

In the section when the TOA method is explanation should be starting with statement that this method is based on knowing the exact time that a signal was sent from the target, the exact time the signal arrives at a reference point, and the speed at which the signal travels. The distance from the reference point can be calculated using the simple equation

distance = c*(tarrival – tsent)

when c is the speed of light. (according [1] )

this explanation should be put before equation (1) row 110 in the article text.

[1]  Shi, G., & Ming, Y. (2016). Survey of Indoor Positioning Systems Based on Ultra-wideband (UWB)Technology. In Wireless Communications, Networking and Applications (pp. 1269-1278). Springer India.

Author Response

Dear Reviewer,

On behalf of my co-authors, we thank you very much for giving us an opportunity to revise our manuscript. We appreciate you very much for your constructive comments and suggestions on our manuscript. Those comments are all valuable and very helpful for improving our manuscript. According to the suggestions and comments, we have tried our best to revise the paper. The English language and writing style are also corrected. We hope these revisions could meet with the approval of the editors and reviewers. All the revisions were highlighted in red in the manuscript. The details of the revisions in the manuscript are explained point by point as follows:

Point 1: In the section when the TOA method is explanation should be starting with statement that this method is based on knowing the exact time that a signal was sent from the target, the exact time the signal arrives at a reference point, and the speed at which the signal travels. The distance from the reference point can be calculated using the simple equation

distance = c*(tarrival – tsent)

when c is the speed of light. (according [1] )

this explanation should be put before equation (1) row 110 in the article text.

[1]  Shi, G., & Ming, Y. (2016). Survey of Indoor Positioning Systems Based on Ultra-wideband (UWB)Technology. In Wireless Communications, Networking and Applications (pp. 1269-1278). Springer India.

Response 1: Thank you for your suggestion. We have already added the explanation and the equation according to your suggestion, referring to lines 129 to l37.

Yours Sincerely,

Yi Jiang

Reviewer 2 Report

This paper focuses on a positioning method based on the joint use of a single shore station and four antennas placed on the ship. The proposed method is quite interesting since in several practical cases the number of AIS shore station is not enough to guarantee an accurate positioning of the vessel. However, the paper presents remarkable flaws, starting from the writing style and the English language. Moreover, it presents poor and unsound results obtained through experiments that are far away from the reality. In fact, the experiments are set on very short distances (200 meters or 0.1 nautical mile), meaning that the ship will be able to use this method only if attached to the shore station. Although these unrealistic scenarios, the results still present not negligible errors, which could be filtered by a not cited or explained filtering procedure. In addition, the methodology is not well described, as well as the reason why this poor accuracy occurs. Here follow some comments on the most critical parts.

In the abstract (lines 16-18) something should have been said on the adopted positioning method and the sentence ‘singular problems due to the limited […]’ is not clear. In line 20 you talk about a discussion on the influencing factors: I cannot find a proper discussion on the motivation which led to these results. In line 36, it seems that only in the AIS R-Mode the AIS shore stations transmit VHF signals. In line 53 and 57, a short explanation on the adopted positioning estimation procedure should have been written. The paragraph from line 60 to 67 seems to be redundant. In line 69, the employed method should have been specified. Paragraph 71-73 because it is not fully understandable. Paragraph 74-76 presents a negative result, that in this case could not be associated to a contribution. Line 91. The figure only shows the TOA method, even if you also introduce the TDOA one. Line 94. A better explanation is missing. Line 120. Figure 2 presents both the local and the global local frame. The figure should consider this, especially because it shows O as the origin of the global one. Line 141. What do you mean with “singularity easily and difficult to solve”? Line 143. The statement “not suitable to solve…” has no references, and the Chan method is not described at all. Line 226. A smooth filtering method is cited, but nothing is written about it. Line 230. How did you choose the scenarios parameters? Moreover, the statement on the number of positioning is not clear. Line 240. You talk about DOP, but there are no results and considerations related to it. The first scenario is represented by a confusing figure: maybe it could be helpful to insert the real position of the vessel. Line 264-266. This statement should have been better explained, maybe with a supporting image. In line 269 you talk again about a filter without giving details. In line 283, a more detailed explanation on this singularity is needed. In line 298-299, RMSE are almost the same with respect to what? Moreover, the substantial differences between the previous scenario are not clear. A major elaboration is missing in the conclusion part, deepening the reason which led to the results.

The figures descriptions are not clear, neither in the caption nor in the layout; measures units, panels, etc. are missing. A little bit of confusion is highlighted in the equations.

Author Response

Dear Reviewer,

On behalf of my co-authors, we thank you very much for giving us an opportunity to revise our manuscript. We appreciate you very much for your constructive comments and suggestions on our manuscript. Those comments are all valuable and very helpful for improving our manuscript. According to the suggestions and comments, we have tried our best to revise the paper. The English language and writing style are also corrected. We hope these revisions could meet with the approval of the editors and reviewers. All the revisions were highlighted in red in the manuscript. The details of the revisions in the manuscript are explained point by point in the attachment.

Yours Sincerely,

Yi Jiang

Round 2

Reviewer 2 Report

Dear authors,

the revisions you made are in line with my suggestions, making the paper more readable and understandable. However, I still notice a lack of details in some sections, so I would suggest making some more improvements which will make the paper more consistent.

First of all, the English should be checked in various section: there are some repetitions and some sentences are not clear. In the following list you can find some changes I would make:

  • In the abstract, you should specify that your method is helpful to solve positioning issue in proximity of the coast
  • Lines 17-21 need to be checked from an English point of view.
  • Check English in line 35, 39-40, 107-109
  • Figure 1 needs to be better described, in line with what you wrote in the text. You can write something like “Coverage areas of AIS stations: different cases”.
  • Line 116: single shore, not only.
  • Check English for lines 166-167
  • Line 267: the antennas position is linked with the ship’s dimensions? Maybe you can clarify this statement
  • You talk about the Monte Carlo method, but neither references nor brief explanation are provided: I suggest adding something.
  • Is figure 4 representing the vessel (as wrote in the description) or the antenna (line 239)? It should be better explained both in the capture and in the text.
  • Lines 301-303: I suggest to insert a comment related to the increase of the RMSE with the heading angle.
  • Line 306: this can be a misunderstanding, but I still cannot understand how you made this simulation. Does the vessel navigate following its route, stop every 5 degrees to stabilize the heading angle on 0 deg and get the position, then continue to navigate?
  • Figure 5 could be bigger to help the visualization of the diamonds. Moreover, I suggest to use the Matlab “legend” function to provide a more intuitive association between colours and image objects.
  • Lines 313-314. What happens when the relative azimuth angle is 0 or 90 degrees?
  • Lines 320-322. English check (missing verbs and words repetitions). You can merge and better explain the link between the matrix and the noise impact on the positioning.
  • Lines 323-324. Why are the RMSEs so different? You could add some details.
  • In line 343 you wrote about the relative position between antenna and station: you can add this consideration in the previous scenario too.
  • Line 349. As in scenario 3, does the ship navigates with a 45 deg route (or similar) and sop every 100 meters to check its position, but only after the heading stabilization?
  • Lines 356-358. Something similar to this statement should be added in the abstract or introduction. However, I suggest to write it better.
  • In the conclusions part you should add some future works, something on how to reduce the errors and in general to improve the method.

Author Response

Dear Reviewer,

On behalf of my co-authors, we thank you very much for giving us an opportunity to revise our manuscript again. We appreciate you very much for your constructive comments and suggestions on our manuscript. Those comments are all valuable and very helpful for improving our manuscript. According to the suggestions and comments, we have tried our best to revise the manuscript. We hope these revisions could meet your approval. All the revisions were marked up using the "Track Changes" of Microsoft Word in the manuscript. The details of the revisions in the manuscript are explained point by point in the attachment.

Yours Sincerely,

Yi Jiang
